# A Novel C@Fe@Cu Nanocomposite Loaded with Doxorubicin Tailored for the Treatment of Hepatocellular Carcinoma

**DOI:** 10.3390/pharmaceutics14091845

**Published:** 2022-09-01

**Authors:** Mohammed S. Saddik, Mahmoud M. A. Elsayed, Amany A. Abdel-Rheem, Mohamed A. El-Mokhtar, Eisa S. Mosa, Mostafa F. Al-Hakkani, Samah A. Al-Shelkamy, Ali Khames, Mohamed A. Daha, Jelan A. Abdel-Aleem

**Affiliations:** 1Department of Pharmaceutics and Clinical Pharmacy, Faculty of Pharmacy, Sohag University, Sohag 82524, Egypt; 2Department of Medical Microbiology and Immunology, Faculty of Medicine, Assiut University, Assiut 71515, Egypt; 3Mining, Metallurgy and Petroleum Engineering Department, Al-Azhar University, Nasr City, Cairo 11371, Egypt; 4Department of Chemistry, Faculty of Science, Al-Azhar University, Assiut Branch, Assiut 71524, Egypt or; 5Department of Physics, Faculty of Science, New Valley University, El-Kharja 72511, Egypt or; 6Department of Pharmacology and Toxicology, Faculty of Pharmacy, Sohag University, Sohag 82524, Egypt; 7Production Engineering Department, Faculty of Engineering, Alexandria University, El-Chatby, Alexandria 21544, Egypt; 8Department of Industrial Pharmacy, Faculty of Pharmacy, Assiut University, Assiut 71515, Egypt

**Keywords:** C@Fe@Cu nanocomposite, doxorubicin, liver cancer, hepatocellular carcinoma

## Abstract

High mortality and morbidity rates are related to hepatocellular carcinoma (HCC), which is the most prevalent type of liver cancer. A new vision for cancer treatment and cancer cell targeting has emerged with the application of nanotechnology, which reduces the systemic toxicity and adverse effects of chemotherapy medications while increasing their effectiveness. It was the goal of the proposed work to create and investigate an anticancer C@Fe@Cu nanocomposite (NC) loaded with Doxorubicin (DOX) for the treatment of HCC. Scanning and transmission electron microscopes (SEM and TEM) were used to examine the morphology of the produced NC. The formulation variables (DOX content, C@Fe@Cu NC weight, and stirring speed) were analyzed and optimized using Box-Behnken Design (BBD) and Response Surface Methodology (RSM). Additionally, X-ray diffraction patterns (XRD) and Fourier Transform Infrared (FTIR) were investigated. Doxorubicin and DOX- loaded C@Fe@Cu NC (DOX-C@Fe@Cu NC) were also assessed against HEPG2 cells for anticancer efficacy (Hepatic cancer cell line). The results revealed the formation of C@Fe@Cu NC with a mean size of 7.8 nm. A D-R model with a mean size of 24.1 nm best fits the adsorption behavior of DOX onto the C@Fe@Cu NC surface. DOX-C@Fe@Cu NC has also been demonstrated to have a considerably lower IC50 and higher cytotoxicity than DOX alone in an in vitro investigation. Therefore, DOX-C@Fe@Cu NC is a promising DOX delivery vehicle for the full recovery of HCC.

## 1. Introduction

Liver cancer is the most common malignancy linked to high mortality and morbidity, with over 90% of all cases being hepatocellular carcinomas (HCCs) [1]. It is possible that a viral infection, obesity, smoking, or drinking led to its emergence, or that it developed as a secondary malignancy [2]. HCC is responsible for 9.1% of all cancer-related deaths. Targeted treatment methods, such as chemotherapy and immunotherapy, have been developed, but their effectiveness has been criticized, and there have been numerous side effects reported as well [1]. Recently, the application of nanotechnology to enhance the efficacy of chemotherapeutic therapies has provided a new perspective in the treatment and targeting of cancer cells with lower tiers of systemic toxicity and fewer serious health consequences [2,3].

One of the most effective anticancer drugs is DOX [4]. Tumors such as hepatic, testicular, lung, and ovarian malignancies are treated with DOX [5], as are leukemia, lymphoma, breast cancer, and other hematological malignancies [6,7]. Furthermore, DOX has been proven to be the first-choice anticancer therapy for hepatic cell carcinoma (HCC) using trans-arterial chemoembolization (TACE) [8]. Intercalation with DNA strands and inhibition of the enzyme topoisomerase 2 are the primary mechanisms by which it exhibits anticancer effects [9]. In addition to this, the metabolic process that it undergoes produces a large number of free radicals [10]. Considering that, its clinical use is still restricted due to acquired resistance, cardiotoxicity, nephrotoxicity, and testicular toxicity [10]. Thus, there is an urgent need for a new formulation design for DOX that improves DOX cell penetration and uptake while avoiding the bad effects of DOX.

In a handful of publications, metallic nanoparticles are demonstrated to be efficacious in the management and targeting of cancer cells. The anticancer capabilities of metal nanoparticles, such as iron, silver (Ag), zinc (Zn), gold (Au), copper (Cu), chromium (Cr), or titanium, can be due to their inherent properties or to surface modifications [10]. It has been revealed that metallic nanoparticles can be exploited as both an active cancer treatment agent and a delivery platform for chemotherapy [1,11].

Herein, we demonstrated for the first time the preparation of C@Fe@Cu NC, which consists of two different types of metallic nanoparticles, iron and copper, as a carrier for DOX. Doxorubicin was successively adsorbed onto the surface of the prepared C@Fe@Cu NC to enhance its anticancer activity. To our knowledge, this is the first study to report the development and use of DOX-C@Fe@Cu NC for the treatment of HCC. Additionally, the morphology, FTIR, and XRD of the produced NC were assessed, and the in vitro anti-tumor activity of DOX-C@Fe@Cu NC against the HepG2 hepatic cancer cell line was studied.

## 2. Materials and Methods

### 2.1. Materials

The lactulose was provided by Sedico company as a free gift (Sedico, Egypt). From Sigma-Aldrich, we purchased copper sulphate pentahydrate CuSO_4_.5H_2_O with a molecular mass of 249.69 g/mol and ferrous sulphate heptahydrate FeSO_4_.7H_2_O with a molecular mass of 278.01 g/mol. Both of these chemicals are sulphates (St Louis, MO, USA). Polyethylene glycols 6000, was obtained from Merck, Germany. As an adsorbate, DOX HCL was purchased from Sigma-Aldrich.

### 2.2. Methods

#### 2.2.1. C@Fe@Cu Nanocomposite Synthesis

Twenty grams of lactulose and fifty grams of polyethylene glycol 6000 (PEG 6000) were dissolved in 100 mL of deionized water, and the pH of the solution was adjusted to an alkaline medium of 12.0 and heated to 70 °C. Copper sulphate pentahydrate with a weight-to-volume of 25.0% and ferrous heptahydrate 2.5% dissolved in deionized water in a 10:1 ratio were mixed together and heated to 70 °C. The mixture solution of copper and ferrous precursors was added slowly to the lactulose solution in a ratio of 1:1, and heated to 70 °C for 60 min with constant stirring at 1000 rpm. Then, the suspension was kept for one day (24 h) to settle. Subsequently, centrifugation was implemented for the suspension at 8000 rpm for 5 min three times. After that, the precipitate was washed using deionized water three times at 8000 rpm via centrifuge. Finally, the separated precipitate was calcinated at 200 °C in the oven for two hours.

#### 2.2.2. Adsorption of DOX by C@Fe@Cu NC

A stock aqueous solution of DOX at a concentration of 10 mg/mL was produced at room temperature (293 K) followed by three new sets of fresh serial dilution concentrations ranging from 20 to 100 µg/mL (F_1_). Two-hour stirring at 300–700 rpm (F_3_) was used to mix 50, 100, and 150 mg of C@Fe@Cu NC (F_2_) with 100 mL of each diluted DOX solution in separate 200 mL beakers. For the adsorption isotherm study, we used 50 mg of C@Fe@Cu NC and 500 rpm stirring speed. After the appropriate amount of time had passed, the Buchner filtration system was used to filter the sample using 0.45 m nylon filter paper. Finally, a spectrophotometer at 480 nm was used to measure the absorbance of each concentration with and without the addition of C@Fe@CuNC as an adsorbent. Accuracy and precision validation standards dictated that all measurements be carried out in triplicates for maximum accuracy and precision [12]. The loaded amount (La) and loading efficiency (LE%) of DOX via C@Fe@Cu NC could be determined according to the following simple equations [13,14,15]:*DOX Le%* = (*Cn* − *Cw*)/*Cn* × 100 (1)
*La* = (*Cn* − *Cw*) × *V/M*
(2)
where *Cn* and *Cw* are the concentrations of DOX without and with C@Fe@Cu NC application in mg/L, respectively.

*V* denotes the DOX experimental volume in *L*, *M* is the applied adsorbent mass (C@Fe@Cu NC) in mg, and *La* denotes the loaded amount of DOX at the surface of C@Fe@Cu NC in mg/g.

To determine the most suitable adsorption model that can be used for the description of the observed adsorption mechanism, four isotherm models were investigated: Langmuir, Freundlich, Temkin, and Dubinin–Radushkevich (D-R) [13,15]. Each model could be tested according to its dedicated mathematical equations as the following [16,17,18]:*Cw*/*La* = (1/*LL KL*) + (1/*LL*) *Cw* -------------- *Langmuir*
(3)
where *LL* is the monolayer loaded capacity of DOX at C@Fe@Cu NC surface (mg/g). In terms of adsorption, *KL* is the Langmuir energy constant (in L/mg).
*RL*= 1/(1 + *KLCmax*) (4)
where *RL* is the separation factor and Cmax is the highest concentration of DOX in the solution at the start (mg/L) without adding C@Fe@Cu NC.
*log La* = *log KF* + (1/*n*) *log Cw* -------------- *Freundlich*
(5)
where, *KF*: Freundlich loaded adsorption capacity of DOX at C@Fe@Cu NC (mg/g). *n*: Freundlich adsorption intensity constant.
*La* = *FT ln BT* + *FT ln Cw* -------------- *Temkin*
(6)
where, *BT* is the binding constant (L/mg), *FT* is Temkin adsorption constant (KJ/mol).
*ln La* = *ln LD-R* − *β* (*RT* (1 + 1/ *Cw*))2 -------------- *D-R*
(7)

*LD-R* is the *D-R* adsorption loaded capacity of DOX at C@Fe@Cu NC in mg/g, *R* is the gas constant (8.314 J/mol K), and *T* is the absolute temperature at 293 K.
*ε* = (*RT* (1 + 1/*Cw*)) (8)
*ED-R* = (−2 *β*) − 1/2 (9)

*ED-R* is the adsorption energy per adsorbed molecule of the DOX at the surface of the C@Fe@Cu NC (kJ/mol).

#### 2.2.3. Characterization of the Prepared Nanocomposite

##### XRD Analysis

At two thetas in the scan range of 4–100 degrees, using a Philips X-ray diffractometer at =1.54056 A “Cu,” we conducted an X-ray powder diffraction powder study (PW 1710, anode material Cu, at a voltage of 40 kV, current of 30 mA, optics: automatic divergence slit, beta filtering using graphite, monochromator). XRD analysis can all be used to investigate crystallographic systems, crystallinity nature, purity, and crystallite size determination [19]. Debye-equation Scherrer’s can be used to express the average crystallite size as follows:Dscherrer = 0.9λ/βhkl cos(θhkl) (10)
where Dscherrer is the average crystallite size, λ is the wavelength for the used x-radiation source which equals = 0.1541838 nm, and βhkl is the corrected full widths at half maxima of the measured peaks, θhkl is the Bragg’s angle diffraction [20,21,22].

##### TEM and SEM Analysis

The particle size and surface morphologies of the NC were investigated using transmission electron microscopy (JEOL model: JEM-100 CXII., Tokyo, Japan),and scanning electron (JEOL model, Tokyo, Japan: JSM 5400LV) [23].

##### Spectroscopic Analysis

FTIR analysis was identified via Thermo Fisher (Nicolet iS10 FTIR Spectrometer, Thermo Fisher Scientific, Waltham, MA, USA) in a wavenumber range of 500–4000 cm^−1^. UV-vis absorption measurements of the DOX adsorption at the surface of C@Fe@Cu NC were recorded at 480 nm using a PerkinElmer [LAMBDA 40] Spectrophotometer using a quartz cell of 1 cm path length at room temperature [23,24].

#### 2.2.4. In Vitro DOX-C@Fe@Cu NC Release Study

At 37 °C, a paddle-type dissolve test device, SR II, with 6 flasks (Hanson Research Co., Chatsworth, CA, USA), operated at 50 rpm, was used to conduct an in vitro release of DOX from the produced C@Fe@Cu NC. 5 mL of 7 M KH_2_PO_4_ containing 16.75 percent (*w*/*v*) NaOH was added in the second hour of the experiment to raise the pH from 1.2 to 7.4, and the experiment continued for another two hours, as previously stated.

It was found that a similar technique was utilized to detect DOX leakage from the C@Fe@Cu NC at a pH of 7.4. An aliquot of three milliliters was aspirated and filtered once every 30 min for measuring the media’s absorbance at its predetermined λ_max_ in comparison with a blank.

##### Design of The Experiment (DoE)

The formulation parameters of the C@Fe@Cu NC composition were assessed and optimized using a Box-Behnken experimental design (BBD) for the maximum LE % and fast drug delivery after one and three hours [25,26,27,28]. Using this design reduces the total number of treatment combinations needed in research involving more than two dependent variables [29,30,31]. Statistical “missing corners” in the BBD model may help the experimenter avoid combined factor extremes when using the model [25,30,32,33]. This functionality ensures that no data will be lost. The DOX-C@Fe@CuNCs were constructed with a three-factor, three-level DoE [34,35,36,37]. The three independent formulation variables studied in this work were DOX concentration, C@Fe@Cu NC weight, and stirring speed. This design was used to construct 15 different DOX-C@Fe@Cu NCs formulas. In the above design, the values −1, 0 and +1 were represented by doses of DOX of 20, 60, and 100 µg/mL. For the C@Fe@Cu NC, the weight was adjusted between 50, 100, and 150 mg; this was labeled as −1, 0, and +1, as well. Finally, a stirring speed of 300, 500, and 700 rpm was chosen to represent a −1, 0, and +1 value. It was decided to evaluate the prepared C@Fe@Cu NCs for loading efficiency (Y_1_), release after one hour (Y_2_), and release after three hours (Y_3_) as the dependent variables.

#### 2.2.5. Effect on Cell Proliferation

##### Cell Line and IC50 Determination

Nawah Scientific Inc. (Cairo, Egypt) supplied the HEPG2 cell line (Hepatic cancer cell line). Supplemented with 10% FBS and 0.1% penicillin-streptomycin, DMEM culture media (Gibco) was used to grow the cells. Cells were incubated in humid air at 37 °C with 5% CO_2_ in all tests.

HEPG2 cells were grown at a density of 1 × 10^4^ cells per well and treated with various concentrations of DOX and DOX-C@Fe@Cu NC ranging from 0.04 to 80 µg/mL to examine the effect on cell survival. Control wells that contained standard growth media were also a subject of the study. After 48 h, the treatment medium was carefully removed and exchanged with PBS. Then, an MTT reagent (Sigma-Aldrich) was added to the cells, and the plate was incubated for a further 4 h. Crystals of formazan were dissolved in DMSO, and the absorbance of the solutions was measured with an Epoch microplate reader at 570 nm (Bio-tek instruments, Santa Clara, CA, USA).

The anti-proliferative effect of drugs was expressed by calculating the relative viability and expressed as a percentage of viability. The viability of cells in the negative control was assumed to be 100%. Viability percentage = absorbance of treated samples divided by absorbance of untreated controls multiplied by 100. Each concentration was tested in triplicates, and the mean was calculated. The IC50 (concentration that leads to 50% inhibition of cellular viability) was determined.

##### Assay for Flow Cytometry

The induction of cell death by DOX was measured with a 7-AAD FITC Annexin V Apoptosis detection kit (Biolegend, San Diego, CA, USA). Briefly, HEPG2 cells were cultured in complete DMEM growth media and treated with DOX, DOX-C@Fe@Cu NC, and normal growth media and incubated for 24 h. PBS was used to clean the cells, Annexin V binding buffer was used to resuspend them, and finally, FITC, Annexin V, and 7AAD were added. Flow cytometry analysis was performed using the FACSCalibur™ flow cytometer (BD Biosciences, San Jose, CA, USA), and FlowJo software 8.7 to analyze the results (Treestar, Ashland, OR, USA).

## 3. Results and Discussion

### 3.1. XRD Analysis

XRD analysis is the best and fastest tool for material identification according to its standard indexed planes [15,38,39,40]. There is a tip peak at 2 θhkl = 11° with d-spacing = 0.8043 nm, which conforms to a graphitic content at hkl Miller plane index (001) for carbon skeleton [41]. The existence of this peak was attributed to an organic content that resulted after NC calcination at 350 °C [15,39,40]. XRD showed mixture planes that were dedicated to the iron and copper nanoparticles according to the reference cards ICDD # 00-431-3213, and 00-901-2043, respectively, of the face center cubic crystallographic system [38]. The reflection peaks (2θhkl) were determined as 22.6°, 23.6°, 25.6°, 31.8°, 34.0°, 37.8°, 46.2°, 52.2°, 80.2°, and 86.8° for iron that identically corresponded to hkl Miller indices of 200, 004, 022, 220, 222, 400, 240, 0410, and 448. However, the reflection peaks (2θhkl) that distinguished copper were 43.2°, 50.4°, 74.2°, 89.8°, and 95.4° with Miller indices of 111, 200, 220, 311, and 222, which were agreed with the previously reported approaches [12,38,42,43,44]. Figure 1 shows that the main peaks were sharp, indicating the high crystallinity of the formed NC [15,40,45]. The diffractogram demonstrated no interference peaks, except for the dedicated peaks to the main components of the NC, confirming the purity of any other oxides [38,40,42]. According to the intensities, the main component of the NC was copper, where carbon and iron acted as a cover coating that prevented copper from oxidation [38,40,42,43,44,46]. Thus, the formed NC could be identified as a C@Fe@Cu nanosystem. The average crystallite size (Dscherrer) was estimated, and it was found to be 12.5 nm. Table 1 shows the detailed XRD calculations of the formed C@Fe@Cu NC.

### 3.2. FTIR Analysis

FTIR analysis is a very important tool to prove the presence of the constituents of active functional groups that are the basic components of any material [43,47,48]. Moreover, FTIR is considered a strong tool for confirming encapsulation or capping of the adsorbent via the adsorbate constituents [14,46]. Figure 2 shows that the Lactulose and C@Fe@Cu NC spectra were shared in most of the lactulose functional groups that appeared in the formed NC with small deviations, especially at stretching bands −OH (3349 cm^−1^), −CH (2926 cm^−1^), −C = O(1617 cm^−1^), and bending band −OH (1070 cm^−1^) [13,42]. It was believed that the −OH and −C = O function groups are responsible for the bio-reduction process of the precursor solutions to the zero state of Cu and Feas different approaches were reported [15,38,39,40,43,46]. Moreover, the sources of carbon content as (shown in XRD) may be attributed to the presence of −CH and −C = O constituents. The absence of the Cu-O (588 cm^−1^, 534 cm^−1^) and Cu_2_O (614 cm^−1^) bands, as well as the Fe-O (521 cm^−1^) bands, indicated that the NC synthesized was a pure Fe and Cu mixture [38,40,49].

The spectra of DOX and DOX-loaded C@Fe@Cu NC show that most of the function groups, particularly those found in the DOX’s fingerprint area (500–1600 cm^−1^) have appeared at the DOX-loaded C@Fe@Cu NC. This finding is solid evidence and effulgent proof of the adsorption process that proceeded at the surface of the C@Fe@Cu NC.

### 3.3. Analysis of TEM, SEM, and Mapping Morphology

TEM and SEM analysis of the C@Fe@Cu NC before and after DOX loading are shown in Figure 3A,C,D,F. The nanoscale integrity of the produced NC was confirmed by TEM and SEM examinations on particles with a size range of 3.9–48.7 nm [40,43,46], where the particle sizes were found to be 7.8 and 24.1 nm before and after DOX loading, respectively. The images assured the crystallographic system that was determined via XRD analysis, where it matched the cubic (red circles) and spherical shapes. When DOX was added to the mixture, the particle size increased significantly, which may be due to the adsorption process [38,40,43]. Moreover, the crystallinity index before and after DOX loading was calculated, and it was noted that it took the behavior toward the polydispersibility nature as 0.4 to 1.1, respectively [14,38,40]. This could be interpreted as an encapsulation process that acted as a coating of the C@Fe@Cu NC system. This may be attributed to the microporous nature of the formed NC (shown in SEM images) [42,43,44,46].

The monodispersed cubic shapes were investigated without any presence of particle agglomeration before the adsorption process of the DOX, where after loading the DOX (as a coating agent), agglomerated particles were observed. The statistical calculations of the TEM analyses are shown in Table 2. The deposited DOX molecules were shown at the surface of the C@Fe@Cu NC in different colors either in TEM or SEM analysis after DOX loading (Figure 3D,F).

Moreover, the carbon presence was confirmed by mapping morphology analysis. Figure 4 shows the distribution of the main component elements of the formed nanocomposite, as it was suggested as a C, Fe, and Cu skeleton. This finding was in line with XRD and FTIR results.

### 3.4. DOX Adsorption onto the C@Fe@Cu NC Surface

Figure 5 highlights the influence of DOX concentration on loading efficiency. With an increase in DOX concentration, the DOX loading increased from 32.5 to 68.4 mg/g. The adsorbent’s empty microporous sites and DOX concentration strive to improve loading efficiency. TEM and XRD analyses confirmed the smaller particle and crystallite sizes of the NC as the main causes of the high surface area that enabled the increase in the adsorption loading tendency [50]. A higher concentration of DOX led to a rise in adsorbed DOX at the sites of the adsorbent until saturation was reached [14,15,40,42].

The progress in DOX loading will be continued to increase from 40 to 60 mg/L to reach a loading of 104.3 mg/g. After a concentration of 60 mg/L of DOX, the loading progress revealed a relatively constant increase. Nevertheless, a slight loading increase would be observed in the trend after the DOX concentration of 100 mg/L to achieve 126.0 mg/g DOX loading capacity at the C@Fe@Cu NC surface. Thus, according to the DOX loading trend, the optimum initial concentration for DOX loading was 60 mg/L.

Four adsorption isothermal models were investigated to demonstrate the most suitable description for the loading efficiency of the DOX adsorption at the surface of C@Fe@Cu NC. The most common applicable isothermal models that underlie adsorption hypotheses are Langmuir, Freundlich, Temkin, and (D-R), which describe the adsorbent/adsorbate relationship [13,14]. Table 3 shows that the best isothermal model that could be used to explain the loading adsorption nature of DOX at the surface of C@Fe@Cu NC is the D-R model. The predilection for choosing the D-R model comes from the highest value of R2 [15,39,42,43,51,52,53,54,55]. As a starting point, isothermal models can be arranged as follows, based on their capacity to represent the loading process, D-R > Langmuir > Temkin > Freundlich.

The visual observation of a color change of DOX from orange to pink after about 30 min of NC addition confirms the adsorption reaction between the DOX molecules and the C@Fe@Cu NC surface (Figure 6).

The D–R isotherm model (Figure 7) could be used for interpretation of the adsorption loading mechanism depending on the assumption of the microporous structure volume for the adsorbent that should be filled [14,56]. In line with these results, SEM demonstrated that the adsorbent morphology (porous material) was compatible with the D–R isotherm model proposal. The maximum loading (LD-R) of the adsorption capacity was 128.8 mg of DOX per 1000 mg of C@Fe@Cu NC.

Moreover, the D-R isotherm model could identify the adsorption class if it is physical or chemical adsorption by calculating the mean free energy (ED-R). There are two types of adsorption processes: physisorption, which occurs when there is less than 80 kJ/mol, and chemisorption, which occurs when there is more than 80 kJ/mol [13]. ED-R = 350.8 kJ/mol, which means that the adsorption process followed the chemisorption path, as demonstrated by our results [14,15].

According to the Langmuir parameter, RL could indicate the probability of adsorption, where if it was more than zero but less than the unit, the adsorption process is desirable, and this has already been achieved (RL = 0.082). Moreover, the Freundlich isothermal model could be used to indicate the adsorption type if it is favorable or not according to the n value, if it is between 1–10; where it was found (2.1), which confirms the favorability of the adsorption process.

### 3.5. DOX Loading Efficiency on Cu/Fe NC (LE %)

Table 4 lists the measurement results using DOX-C@Fe@Cu NC Formulations 1, 5, and 12 each contained 20 µg/mL of DOX, but the theoretical drug content of formulations 2, 3, 7, 8, 9, 13, or 14 totaled 60 µg/mL or 100 µg/mL of the medication, respectively. Changes in DOX conc., C@Fe@Cu NC weight, and stirring speed were investigated using the ANOVA test to check whether the LE percentage was affected. The range of LE percent values was 45.87 ± 2.03 to 82.11 ± 3.23 and was dependent on DOX concentration, C@Fe@Cu NC weight, and stirring speed. Multiple linear regression models are used to describe the relationship between DOX concentration and LE %. In the model that was fitted, the equation states:LE % = 69.45 + 7.21 F_1_ + 6.45 F_2_ − 4.99 F_3_ − 1.19 F_1_^2^ − 5.24 F_2_^2^ + 2.25 F_3_^2^ − 1.31 F_1_F_2_ + 0.57 F_1_F_3_ − 0.14 F_2_F_3_
(11)

Using the R-squared statistic, we can observe that the model as fitted accounts for 93.96% of the variance in DOX conc. A multiple linear regression model was fitted to the data, and the output reveals how the weight of C@Fe@Cu NCs relates to the LE %. At a 95% level of confidence, the ANOVA table shows that the variables exhibit a statistically real correlation. As long as the *p*-value is greater than 0.05, stirring speed has a significant impact on LE%.

Using a 3-D plot (Figure 8), LE percentage increased from 45.87 ± 2.03 to 64.32 ± 2.08 and from 70.25 ± 2.21 to 82.11 ± 3.23 at lower and higher DOX concentrations with a constant C@Fe@Cu NC weight and stirring speed. Higher DOX concentrations promote drug adsorption and incorporation on the surface of C@Fe@Cu NC, leading to higher LE and DOX conc. After forming stable C@Fe@Cu NC, DOX can be adsorbed onto the surface of C@Fe@Cu NC in the second main step. The LE percent of nanocomposites produced is significantly influenced by these two stages. Similar results were observed when biodegradable sorafenib was put into carbon nanotubes to treat hepatocellular cancer [1].

On the other hand, LE% increased from 65.64 ± 3.98 to 75.89 ± 2.01% and 57.31 ± 2.23 to 66.81 ± 3.43% to varying levels, lower and higher C@Fe@Cu NC weight with a constant DOX conc., and stirring speed (Table 4). The surface area accessible for drug loading increased as the weight of C@Fe@Cu NC rose, permitting more medication to be adsorbed on the nanocomposite’ surfaces. The LE percentage increased from 65.64 ± 3.98 percent to 75.89 ± 2.01 percent and from 57.31 ± 2.23 percent to 66.81 ± 3.43 percent, at lower and higher Cu/Fe NCs weight levels, respectively. As the weight of nanocomposites goes up, more DOX can be absorbed on their surface because the area of the surface that can be used to load drugs goes up as well.

While preparing nanocomposites, stirring speeds had a significant impact on the LE%, which dropped as the stirring speed increased. The increased stirring speed during the adsorption process hindered the adsorption of DOX at the surface of C@Fe@Cu NC. The primary goal of the stirring speed investigation was to determine the optimum rotational speed that yielded the highest LE%. At constant DOX level and C@Fe@Cu NC weight, the LE percent declined from 65.64 ± 3.98 to 57.31 ± 2.23% (N2, N3) and from 70.25 ± 2.21 to 57.76 ± 3.05% (N5, N6) at lower and higher temperatures (Table 4). All of these findings demonstrate that you need to stir at a slower speed (300 rpm, −1) to obtain a higher LE%.

### 3.6. In Vitro Release Study of DOX-C@Fe@Cu NC

Results of fitting a multiple linear regression model for the correlation between DOX conc., C@Fe@Cu NC weight, and stirring speed, and % drug release after one or three hours are shown in Equations (12) and (13).
Rel 1 h = 26.10 + 3.365 F_1_ + 2.420 F_2_ − 0.527 F_3_ − 3.38 F_1_^2^ − 2.31 F_2_^2^ + 1.27 F_3_^2^ − 0.21 F_1_F_2_ + 0.19 F_1_F_3_ − 1.18 F_2_F_3_(12)
Rel 3 h = 89.73 + 7.37 F_1_ + 2.27 F_2_ − 1.15 F_3_ − 5.35 F_1_^2^ − 7.47 F_2_^2^ + 2.86 F_3_^2^ − 3.37 F_1_F_2_ + 2.06 F_1_F_3_ − 5.05 F_2_F_3_
(13)

A significant difference in drug release was observed with increasing DOX conc. but not with increasing C@Fe@Cu NC weight after one hour or three hours, since the *p*-value in the ANOVA table is less than 0.05. In contrast to its impact on LE percent, stirring speed had no influence on drug release.

The in vitro release of DOX from its C@Fe@Cu NC is shown in Figure 9 and Figure 10 and Table 4 as response surface plots. At the end of an hour, the percent released was 24.24 ± 1.74 and 34.12 ± 1.77, respectively (Y_2_). After three hours (Y_3_) of dissolution, the in vitro release reached a maximum of 97.73 ± 1.33 percent and a minimum of 67.15 ± 1.96 percent. Thus, based on the amount of drug released in vitro, the formulations can be ranked as follows: N3 > N2 > N4 > N1.

C@Fe@Cu NC featuring formulae N5, 6, 7, 8, 9, 10, and 11 have been used in the in vitro release of DOX from their C@Fe@Cu NC (F_2_) at a medium level (0), varied stirring speed levels (ranging from -1 to +1), and with varying DOX concentration (F_1_). A total of 28.33 ± 1.43 percent and a minimum of 21.89 ± 1.98 percent had been released by the conclusion of the hour (Y_2_). After three hours of dissolution (Y_3_), an in vitro release of a 97.73 ± 1.33 percent and an in vitro release of 81.09 ± 1.94 percent were observed.

At constant DOX concentration and temperature (0, 1), the maximum and minimum in vitro release were 95.77 ± 1.72 percent and 74.93 ± 1.54 percent, respectively, for lower and higher C@Fe@Cu NC weights (N3, N14). After one hour, the percentage released decreased from 31.90 ± 1.65 percent to 25.92 ± 1.99 percent, and after three hours, it decreased from 84.42 ± 1.33 percent to 79.18 ± 1.63 percent, as were similar findings for N4 and N15 at lower and higher C@Fe@Cu NC weights.

All of these results demonstrated that increasing DOX concentrations had a significant impact on the drug’s adsorption within one and three hours. C@Fe@Cu NC adsorbs the drug, making it easier to release, and the amount of drug accessible for release increases as the concentration of the drug increases. When compared to the amount released after three hours, the amounts released after an hour were insignificant. In an acidic environment, DOX desorption from C@Fe@Cu NC is more difficult, and the DOX-C@Fe@Cu NC system is still intact at pH 1.2. DOX and coated C@Fe@Cu NC, which make more medication available for absorption at sites with an alkaline pH, experience rapid desorption behavior at alkaline pH, increasing the amount released considerably. Reducing medication release in the stomach could reduce the occurrence of gastritis and other DOX adverse effects.

The weight of the C@Fe@Cu NC negatively correlated with the percentage of DOX released. With increasing NC weight, the surface area of C@Fe@Cu NC increased. It allowed the monolayer pattern to adsorb more DOX. Drug release from the C@Fe@Cu NC surface is negatively impacted by DOX monolayer adhesion. The medication is adsorbing in a multilayer pattern to the surface of C@Fe@Cu NC, which reduces the attachment forces and, makes it more readily available for release. Stirring speed, in contrast to the LE percent, did not affect DOX release after one and three hours.

In vitro results demonstrated that the ideal formula of DOX-C@Fe@Cu NC should be prepared at a high concentration of DOX (+1, 100 µg/mL), a medium concentration of C@Fe@Cu NC weight (0, 100 µg/mL), and a moderate stirring speed (300 rpm). According to the rank order of formulae, N10 is the best formula in terms of LE percentage and Rel 1 and 3 h.

### 3.7. In Vitro Cytotoxicity Assay

Using the MTT assay, the cytotoxic impact of Dox-C@Fe@Cu NCs on HEPG2 cells was studied and contrasted with that of DOX alone. The cytotoxic effect of DOX and DOX-C@Fe@Cu NC was dose-dependent (Figure 11a). It is interesting to note that DOX-C@Fe@Cu NCs had a lower IC50 (1.47 g/mL) than DOX alone (2.78 g/mL). Flow cytometric analysis of apoptotic cells confirmed our findings, demonstrating that cells treated with DOX-C@Fe@Cu NCs (5 g/mL) had a higher percentage of late apoptotic cells (27.5 percent) than cells treated with DOX at the same dose (17.9 percent) Figure 11b–d. More cancer cells are dying because apoptosis is getting better. This shows that the treatment is affecting the growth of tumor cells.

## 4. Conclusions

At the nanoscale level, the DOX-C@Fe@Cu NC that was constructed is properly sized for drug delivery to the tissues of the tumor. On the LE percent, observable effects of formulation parameters such as DOX concentration, C@Fe@Cu NC weight, and stirring speed were found to be significant. Because it significantly reduced the IC50 of DOX and exerted more cytotoxicity than DOX alone, C@Fe@Cu NC loaded with DOX could be a promising therapy for the treatment of liver cancer. This is because it was encountered that C@Fe@Cu NC loaded with DOX exerted more cytotoxicity than DOX alone, despite the fact that it was employed in lower doses than DOX alone. Because of this, the toxicity of DOX, which is extremely dose-dependent, will be considerably reduced. Based on our interesting findings, it is suggested to look into how DOX-C@Fe@Cu NC could be used to reduce clinical resistance to DOX and its toxic effects at high doses.

## Figures and Tables

**Figure 1 pharmaceutics-14-01845-f001:**
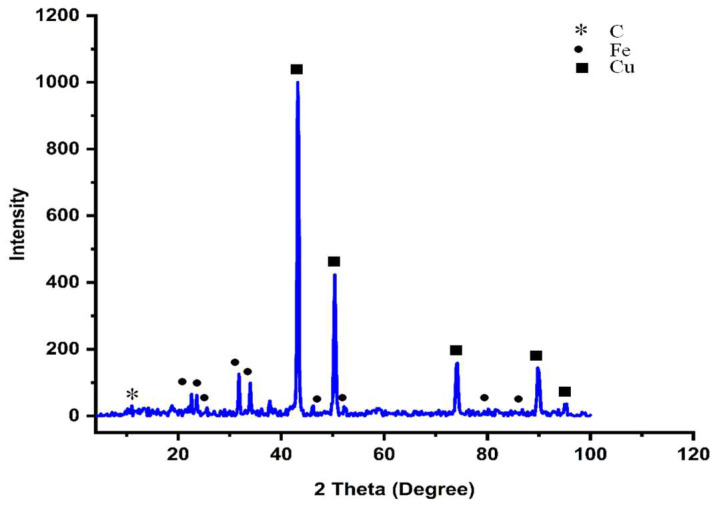
XRD diffractogram of the C@Fe@Cu NC.

**Figure 2 pharmaceutics-14-01845-f002:**
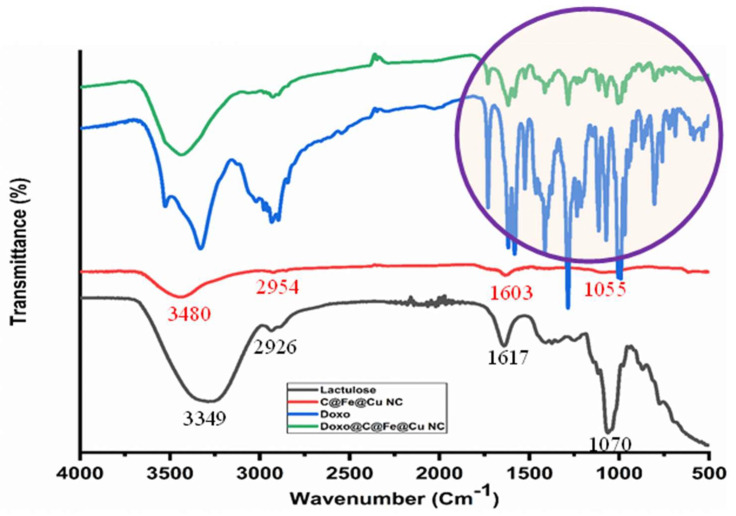
FTIR spectra of Lactulose, C@Fe@Cu NC, DOX, and DOX loaded C@Fe@Cu NC.

**Figure 3 pharmaceutics-14-01845-f003:**
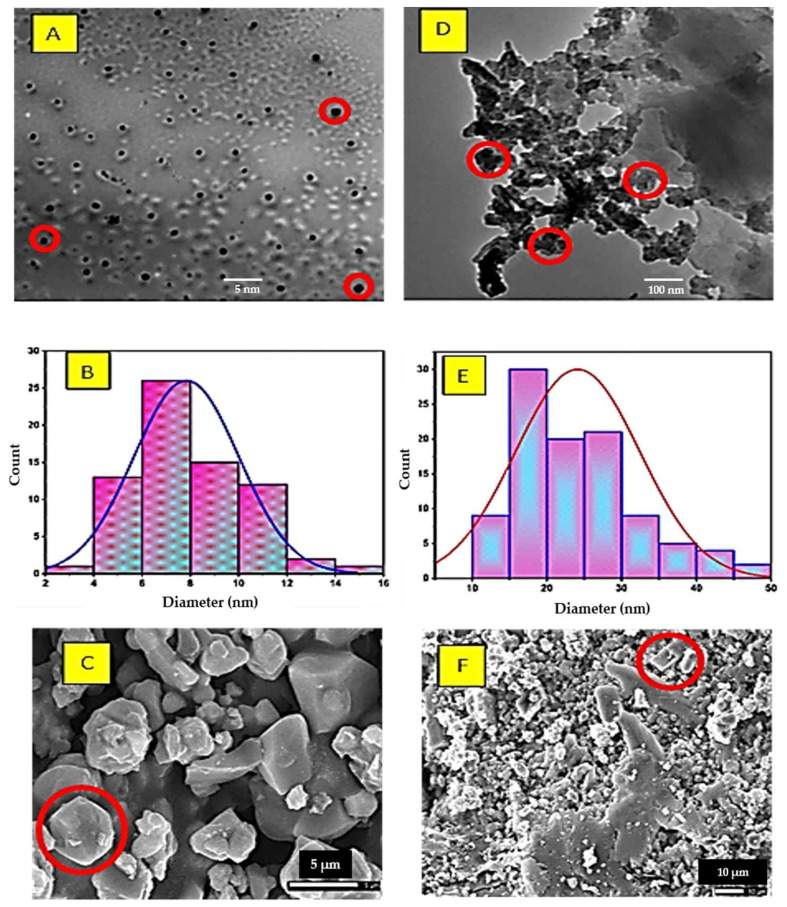
TEM, particle size distribution, and SEM analysis of C@Fe@Cu NC (**A**–**C**) and DOX adsorption by C@Fe@Cu NC (**D**–**F**).

**Figure 4 pharmaceutics-14-01845-f004:**
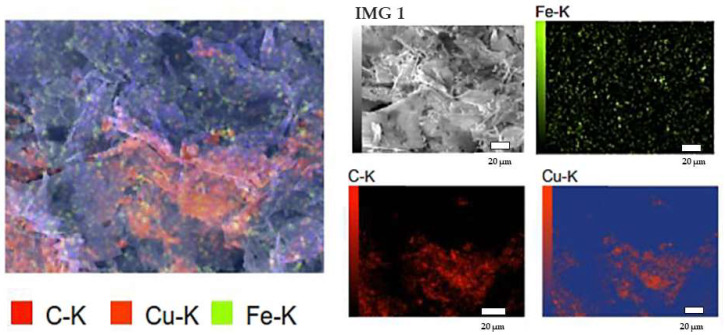
Mapping analysis of C@Fe@Cu NC.

**Figure 5 pharmaceutics-14-01845-f005:**
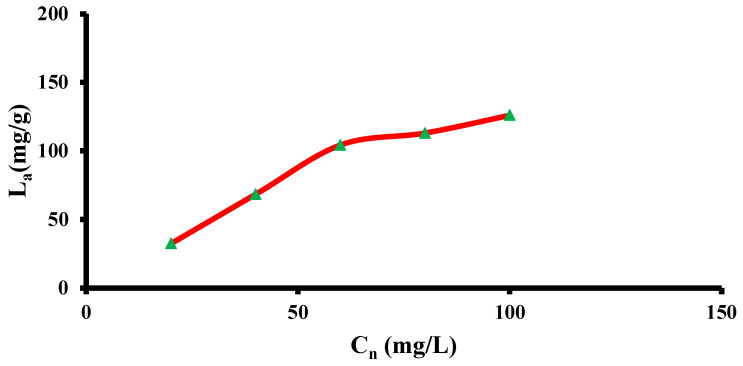
Loading efficiency via C@Fe@Cu NC at different concentrations of DOX.

**Figure 6 pharmaceutics-14-01845-f006:**
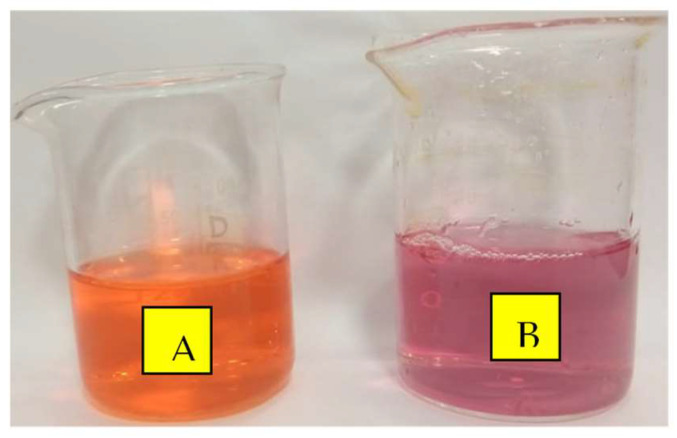
Adsorption color change impact of adsorbed DOX molecules at the C@Fe@Cu NC surface (**A**) before; (**B**) after adsorption.

**Figure 7 pharmaceutics-14-01845-f007:**
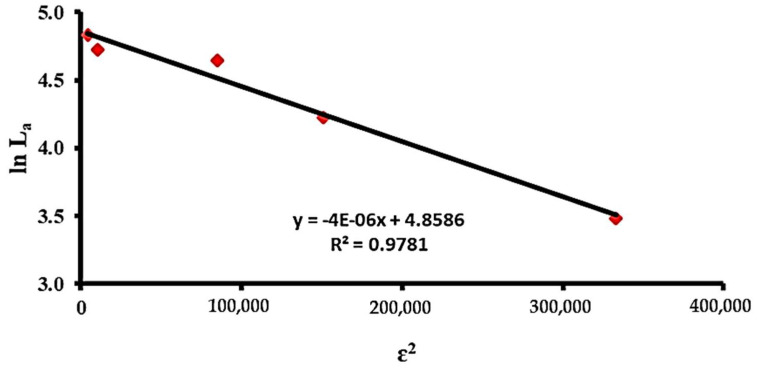
D–R isothermal model of the DOX molecules’ chemical adsorption at the C@Fe@Cu NC surface.

**Figure 8 pharmaceutics-14-01845-f008:**
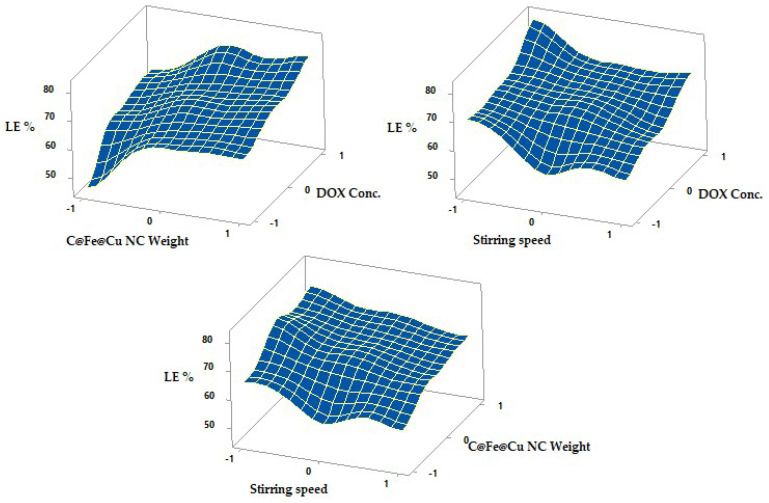
Effects on LE% of different formulation parameters are shown as surface plots.

**Figure 9 pharmaceutics-14-01845-f009:**
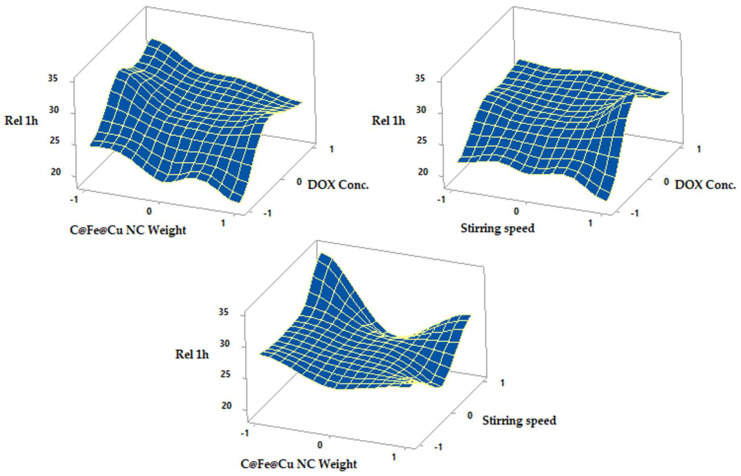
Surface plots showed the effects of the several formulation parameters on release for one hour.

**Figure 10 pharmaceutics-14-01845-f010:**
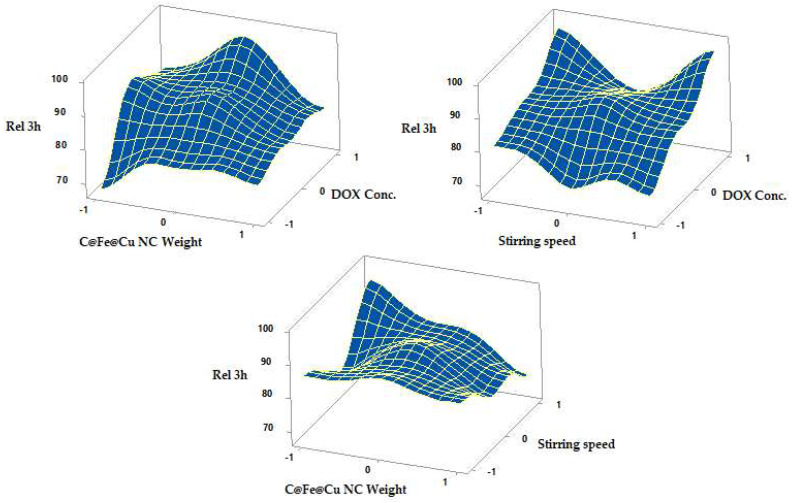
The surface illustrates the influence of different formulation variables on the release after one hour.

**Figure 11 pharmaceutics-14-01845-f011:**
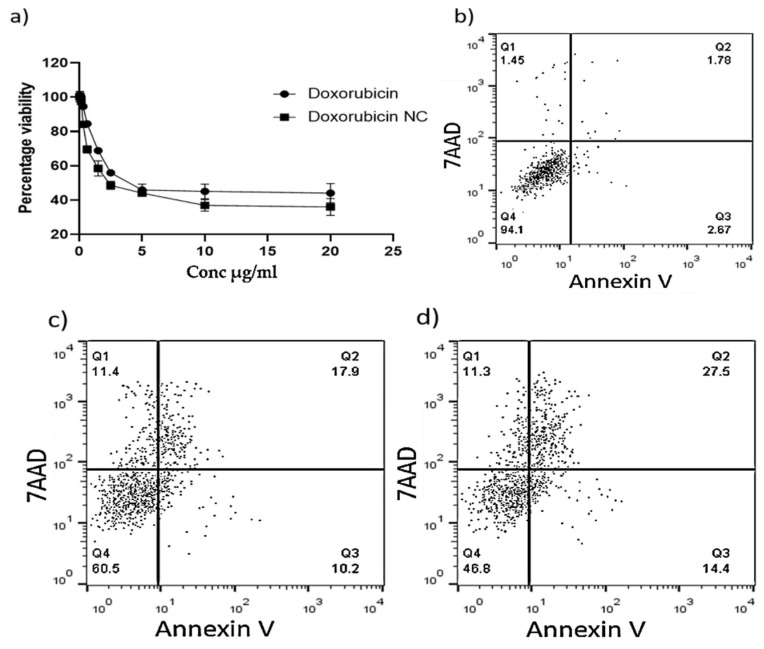
MTT assay demonstrating the viability of HEPG2 cells treated with various doses of DOX and DOX NC (**a**); Analysis of flow cytometry shows how frequently cells treated with control growth media go into apoptosis (**b**); DOX (**c**); and DOX-NC (**d**).

**Table 1 pharmaceutics-14-01845-t001:** The C@Fe@Cu NC XRD measurements.

2θ°_Reference_	System	2θ°_hkl Measured_	Miller Indices			S*_cherrer_* (nm)	d_reference_ (nm)	d_calculated_ (nm)
			*h*	*k*	*ℓ*			
10.6	(C) *	11.0	0	0	1	6.716	0.8352	0.8043
22.6	Fe peaks (fcc) **	22.6	2	0	0	8.214	0.3934	0.3934
23.5		23.6	0	0	4	6.830	0.3782	0.3770
25.6		25.6	0	2	2	10.402	0.3476	0.3480
32.2		31.8	2	2	0	5.944	0.2774	0.2814
34.4		34.0	2	2	2	5.222	0.2605	0.2637
46.1		46.2	4	0	0	14.948	0.1967	0.1965
52.2		52.2	2	4	0	8.969	0.1752	0.1752
80.1		80.2	0	4	10	10.530	0.1197	0.1197
87.1		86.8	4	4	8	7.868	0.1119	0.1122
43.3	Cu peaks (fcc) ***	43.2	1	1	1	21.737	0.2087	0.2094
50.4		50.4	2	0	0	19.618	0.1808	0.1811
74.1		74.2	2	2	0	18.307	0.1278	0.1278
89.9		89.8	3	1	1	20.376	0.1090	0.1092
95.2		95.4	2	2	2	22.292	0.1044	0.1042
Average			------------------------			12.53	-	-

* Noted for carbon planes, ** iron planes, *** copper planes.

**Table 2 pharmaceutics-14-01845-t002:** TEM analysis results of C@Fe@Cu NC before and after DOX loading.

Item	Before	After
Average particle sizes (nm)	7.8	24.1
Standard deviation (nm)	2.2	8.3
Minimum particle size (nm)	3.9	10.2
Maximum particle size (nm)	14.3	48.7
Median (nm)	7.8	23.1

**Table 3 pharmaceutics-14-01845-t003:** Applicable isotherm models of the loading adsorption of DOX at the C@Fe@Cu NC surface.

Item	Isothermal Models			
	Langmuir	Freundlich	Temkin	D-R
R2	0.9352	0.7011	0.8065	0.9781
Model parameter	LL = 158.3	n = 2.1	BT = 1.14	LD-R = 128.8
	kL = 0.111	kF = 25.5	FT = 35.4	β = −4.1 × 10^−6^
	RL = 0.082			ED-R = 350.8

**Table 4 pharmaceutics-14-01845-t004:** Observed values of responses for DOX-loaded C@Fe@Cu NC.

Identifier System No.	Coded Form of the Variable Level			LE%	Cumulative Percent Released	
	F_1_ DOX Conc.	F_2_ C@Fe@Cu NC Weight	F_3_ Stirring Speed	Y_1_ LE%	Y_2_ Rel 1 h	Y_3_ Rel 3 h
N1	−1	−1	0	45.87 ± 2.03	24.88 ± 1.88	67.15 ± 1.96
N2	0	−1	−1	65.64 ± 3.98	28.26 ± 1.93	86.77 ± 1.11
N3	0	−1	1	57.31 ± 2.23	34.12 ± 1.77	95.77 ± 1.72
N4	1	−1	0	64.32 ± 2.08	31.90 ± 1.65	84.42 ± 1.33
N5	−1	0	−1	70.25 ± 2.21	21.89 ± 1.98	81.09 ± 1.94
N6	−1	0	1	57.76 ± 3.05	19.83 ± 1.34	73.13 ± 1.56
N7	0	0	0	64.13 ± 2.06	28.14 ± 1.65	92.11 ± 1.65
N8	0	0	0	73.23 ± 4.12	24.24 ± 1.74	89.51 ± 1.44
N9	0	0	0	70.99 ± 3.11	24.02 ± 1.83	87.93 ± 1.76
N10	1	0	−1	82.11 ± 3.23	28.33 ± 1.43	96.23 ± 1.34
N11	1	0	1	71.91 ± 3.87	26.36 ± 1.87	97.73 ± 1.33
N12	−1	1	0	64.33 ± 2.99	19.88 ± 1.96	76.45 ± 1.93
N13	0	1	−1	75.89 ± 2.01	26.02 ± 1.54	85.69 ± 1.25
N14	0	1	1	66.81 ± 3.43	28.61 ± 1.66	74.93 ± 1.54
N15	1	1	0	77.54 ± 2.02	25.92 ± 1.99	79.18 ± 1.63

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
