# Peer review of "A Novel C@Fe@Cu Nanocomposite Loaded with Doxorubicin Tailored for the Treatment of Hepatocellular Carcinoma"

_pharmaceutics, 2022, doi:10.3390/pharmaceutics14091845_

Round 1
Reviewer 1 Report
|
The paper reveals a very good research regarding drug delivery systems within cancer management. The research is very opportune to enhance the knowledge over drug delivery systems in general. |
The study requires several approached to be addressed.
- The doxorubicin anti-cancer drug is well known for its efficiency against breast cancer. Therefore, please add several references on this cancer direction in Introduction section.
- The role of the lactulose and PEG6000 must be specified.
- One big issues of this research is the huge doxorubicin amount, 100 g/ml (line 102), used for the encapsulation (the required amount should be 1000 times lower- at most 100 mg/ml to be sure that the drug is well dissolved).
Lines 102- 104: You said that the initial solution (100 g/ml) was diluted in the range 20 - 100 g/ml.
If you will use 100 mg of nanoparticles (line 104) for mixing with 50 ml of each diluted DOX solution (line 105), this means that for 100 mg of nanoparticles will be more than 1000 g of doxorubicin (50 ml x 20 g/ml= 1000 g ?????)
This amount is higer with respect to doxorubicin water solubility. Therefore, the nanoparticles are entrapped within a doxorubicin extra mass.
This type of drug delivery system is not suitable to be used within cancer management because the most of the drug amount will behave as a classic system where the drug is unprotected and the sustained/controlled release does not occurs.
- Furthermore, the encapsulation section (2.2.2.) does not match with the DOX loading onto the C@Fe@Cu NC surface section (3.4). There are values about 32.5 mg/g. If there 32.5 mg is the drug amount for every gram of nanoparticles, you will have 3.25 mg of drug for each 100 mg of nanoparticles (disscused above for the 2.2.2 section). In conclusion, you used 1000 g of drug in the encapsulation for 100 mg of nanoparticle......and only 3.25 mg remained?.... this is misleading!!!!
Furthermore, in the 3.4 section are values of 40 to 60 mg/L.....where these values were used before? If these values are real, they must be included in the Methods section (probably 2.2.2).
- A complex doxorubicin release investigation must be included.
Author Response
Dear Editor and Reviewers:
Thank you for your helpful comments concerning our manuscript entitled “A novel C@Fe@Cu Nanocomposite Loaded with Doxorubicin Tailored for the Treatment of Hepatocellular Carcinoma” (pharmaceutics-1850352). These opinions are all valuable and helpful for further revising and improving the quality of the manuscript. We have studied the suggestions carefully and made the corrections accordingly.
The detailed responds to the reviewer’s comments are shown as follows
Response #1: Thank you very much for your suggestions and we really appreciate your efforts
in reviewing our manuscript. We have revised the manuscript accordingly. Our point-by-point
responses are detailed as below.
1 - The doxorubicin anti-cancer drug is well known for its efficiency against breast cancer. Therefore, please add several references on this cancer direction in Introduction section.
Many thanks for this valuable suggestion, we add references 6 and 7 for this cancer direction
2 - The role of the lactulose and PEG6000 must be specified.
Lactulose act as a reducing agent for the bioremediation green synthesis of copper and ferrous precursors. PEG 6000 act as a homogenizing agent for distribution the produced nanocomposite.
3 - One big issues of this research are the huge doxorubicin amount, 100 g/ml (line 102), used for the encapsulation (the required amount should be 1000 times lower- at most 100 mg/ml to be sure that the drug is well dissolved).
Lines 102- 104: You said that the initial solution (100 g/ml) was diluted in the range 20 - 100 g/ml. If you will use 100 mg of nanoparticles (line 104) for mixing with 50 ml of each diluted DOX solution (line 105), this means that for 100 mg of nanoparticles will be more than 1000 g of doxorubicin (50 ml x 20 g/ml= 1000 g ?????). This amount is higher with respect to doxorubicin water solubility. Therefore, the nanoparticles are entrapped within a doxorubicin extra mass.
We apologize for this unintended typing error. We corrected that in the manuscript. The right concentration is mg/mL, not g/mL "a stock aqueous solution of DOX was prepared at a concentration of 10 mg/mL." Dilutions are 20, 60, and 100 µg/ml. We add 100 mL of these dilutions to provide a sufficient amount of DOX to be adsorbed on our NC surface. For the adsorption isotherm study, we used 100 µg/mL of DOX, 50 mg of C@Fe@Cu NC and 500 rpm stirring speed. Section 2.2.2 now includes a new section highlighted in red, as follows: :
A stock aqueous solution of DOX at a concentration of 10 mg/mL was produced at room temperature (293 K) followed by three new sets of fresh serial dilution concentrations ranging from 20 to 100 µg/mL (F1). Two-hour stirring at 300-700 rpm (F3) was used to mix 50, 100, and 150 mg of C@Fe@Cu NC (F2) with 100 mL of each diluted DOX solution in separate 200 mL beakers. For adsorption isotherm study, we used 100 µg/mL of DOX, 50 mg of C@Fe@Cu NC and 500 rpm stirring speed. After the appropriate amount of time had passed, the Buchner filtration system was used to filter the sample using 0.45m nylon filter paper. Finally, a spectrophotometer at 480 nm was used to measure the absorbance of each concentration with and without the addition of C@Fe@Cu NC as an adsorbent. Accuracy and precision validation standards dictated that all measurements be carried out in triplicates for maximum accuracy and precision [13-17]. The loaded amount (La) and loading efficiency (LE%) of DOX via C@Fe@Cu NC could be determined according to the following simple equations
4- This type of drug delivery system is not suitable to be used within cancer management because the most of the drug amount will behave as a classic system where the drug is unprotected and the sustained/controlled release does not occur.
1- Your comment is valuable in evaluating the pharmacokinetics of the formulation but doxorubicin is given twice per week so , it is not significant for us in this study to create a controlled or slow-release formula but we want the formulation to stay as one unit will be taken by hepatic cancer cells where the active drug (doxorubicin) can act.
2-The purpose of our Formulation is to direct the drug to the cancer cells depending on the affinity of the cancer cells for iron and copper more than in normal cells, where copper and iron are very essential elements for cancer cells for proliferation and growth thus increasing the rate of drug uptake by cancer cells more than normal cells (selectivity) so increasing selectivity and decreasing toxicity(Jung et al., 2019) (Hsu et al., 2020).
3- There are many types of research about using adsorption mechanisms of anti-cancer drugs such as (Arora et al., 2012; Cai et al., 2019; Karimzadeh et al., 2021; Khan et al., 2021)
4- Because adsorption is a physical method it will not affect SAR of doxorubicin rather than chemical methods
5 - Furthermore, the encapsulation section (2.2.2.) does not match the DOX loading onto the C@Fe@Cu NC surface section (3.4). There are values of about 32.5 mg/g. If there 32.5 mg is the drug amount for every gram of nanoparticles, you will have 3.25 mg of the drug for each 100 mg of nanoparticles (discussed above for the 2.2.2 section). In conclusion, you used 1000 g of drug in the encapsulation for 100 mg of nanoparticle......and only 3.25 mg remained?.... this is misleading!!!!
We corrected the title in section 3.4 it is an adsorption isotherm study. For the adsorption isotherm study, we used 100 µg/mL of DOX (100 mL), 50 mg of C@Fe@Cu NC, and 500 rpm stirring speed. We think now all calculations are clear, regarding adsorption isotherm and LE%.
6- Furthermore, in the 3.4 section are values of 40 to 60 mg/L.....where these values were used before? If these values are real, they must be included in the Methods section (probably 2.2.2).
Already, these concentrations were included in section 2.2.2 in line 104 “from 20 to 100 µg/mL (F1)…..” Where here we discussed and interpreted the adsorption status profile about the increase or relative increase of the adsorption loading of DOX onto the C@Fe@Cu NC by studying the effect of increasing the initial concentration of the DOX from 20 µg/mL to 100 µg/mL.
7 - A complex doxorubicin release investigation must be included.
- We already investigated drug release at simulated gastrointestinal fluid at different pHs 1.2 and 7.4. Section 3.6 and figures 9 and 10 show the effect of different formulation variables on DOX release after 1 and 3 h.

Reviewer 2 Report
The authors have done good work, done many experiments, and provided good results; however, they have not presented their data very well and readable. As a reader, in most cases, I cannot follow them. I have provided some general comments below. I am more than happy to review the article after you revise it.
- The article should be revised in terms of the English language.
- The authors need to be consistent in writing. I can see, e.g., in some places, they wrote ”anticancer” and in some “anti-cancer.”
- Abbreviations should be defined at first mention, and then the authors need to use the abbreviations. E.g., Doxorubicin has been abbreviated by the authors as DOX, but the authors used DOX, Dox, and Doxorubicin throughout the article!
- I cannot understand the following statement! Can the authors explain it?
“Both solid tumors such as hepatic, testicular, lung, and ovarian malignancies are treated with DOX [5], leukemia, lymphoma, and other hematological malignancies [6].”
- The quality of some images is not good, and I cannot see the numbers on some graphs. They need to be fixed. For example:
Fig. 3 (A-D) The numbers and scale bars are not clear
Fig. 4. The scale bar – this is a paper! 20 um does not mean anything. It must be 20 µm
Fig. 5. Why is the x-axis from 0-150? Why not 0-100?
Fig. 7. The figure needs to be revised. The numbers are on the X-axis.
Fig. 11. The numbers are not clear
- Some of the figures have panels, but in the captions, there is no panel. The captions need to be revised.
- This is just an example. The authors wrote, “PEG 6000, often known as polyethylene glycols 6000, was obtained from … .” PEG 6000 is an abbreviation for polyethylene glycols 6000. What does “often known as polyethylene glycols 6000” mean? This article needs a significant revision in terms of writing.
- Why is the material section bold?
- Some of the numbers do not have the unit. E.g., line 91: 100 mL.
- What are these ratios?
Line 93: 10:1
Line 96: 1:1
- I am not sure why the authors use Kelvin (e.g., line 103) in some places in the article. They should be Celsius.
- I cannot follow the methods and materials sections. The authors should write these sections in detail so that others can repeat the experiments.
- The authors claim that the red circles in Fig. 3 are cubic particles. As a reader, I cannot see any cubic particles specifically in panels A and D.
- The conclusion should be expanded.
Author Response
Dear Editor and Reviewers:
Thank you for your helpful comments concerning our manuscript entitled “A novel C@Fe@Cu Nanocomposite Loaded with Doxorubicin Tailored for the Treatment of Hepatocellular Carcinoma” (pharmaceutics-1850352). These opinions are all valuable and helpful for further revising and improving the quality of the manuscript. We have studied the suggestions carefully and made the corrections accordingly.
The detailed responds to the reviewer’s comments are shown as follows
Reviewer #2: Thank you very much for your suggestions and we really appreciate your efforts
in reviewing our manuscript. We have revised the manuscript accordingly. Our point-by-point
responses are detailed below.
- - The article should be revised in terms of the English language.
- Thank you for your comments on improving this manuscript. We have thoroughly revised the manuscript and checked it with the Grammarly® premium version and cross-checked by a native English speaker.
2- The authors need to be consistent in writing. I can see, e.g., in some places, they wrote ”anticancer” and in some “anti-cancer.”
- Corrected
3- Abbreviations should be defined at first mention, and then the authors need to use the abbreviations. E.g., Doxorubicin has been abbreviated by the authors as DOX, but the authors used DOX, Dox, and Doxorubicin throughout the article!
- Many thanks for this valuable comment, we unify abbreviations throughout the article.
4- I cannot understand the following statement! Can the authors explain it?
“Both solid tumors such as hepatic, testicular, lung, and ovarian malignancies are treated with DOX [5], leukemia, lymphoma, and other hematological malignancies [6].”
We corrected this sentence,
Tumors such as hepatic, testicular, lung, and ovarian malignancies are treated with DOX, as are leukemia, lymphoma, breast cancer, and other hematological malignancies
5- The quality of some images is not good, and I cannot see the numbers on some graphs. They need to be fixed. For example:
We are sorry for the unpleasant reviewing process! We initially uploaded all the figures with a resolution higher than 600 dpi, but they were all compressed in the reviewing manuscript. Now, we again updated all the figures with a higher resolution.
6 - Fig. 3 (A-D) The numbers and scale bars are not clear
We updated the figure with a higher resolution.
7- Fig. 4. The scale bar – this is a paper! 20 um does not mean anything. It must be 20 µm
We are so sorry of that, we corrected it.
8- Fig. 5. Why is the x-axis from 0-150? Why not 0-100?
Many thanks for this suggestion. Just to be more clarified the start and the end resultant data. If we use the 0-100, the reader may be thinking the results were be cut and seemed to be uncompleted.
9- Fig. 7. The figure needs to be revised. The numbers are on the X-axis.
The results were be revised and the values on the X-axis represents the actual calculations according to equation 7-9 of D-R adsorption loaded capacity isothermal model.
10- Figure 11
Corrected
11- - Some of the figures have panels, but in the captions, there is no panel. The captions need to be revised.
- This is just an example. The authors wrote, “PEG 6000, often known as polyethylene glycols 6000, was obtained from … .” PEG 6000 is an abbreviation for polyethylene glycols 6000. What does “often known as polyethylene glycols 6000” mean? This article needs a significant revision in terms of writing.
- All these errors were corrected.
12- Some of the numbers do not have the unit. E.g., line 91: 100 mL.
We apologize for this unintended missing unit. The unit was added, thank you.
13 - What are these ratios?
Line 93: 10:1
Mixture of copper sulphate pentahydrate and ferrous heptahydrate in concentration 27.5 % wt/v dissolved in deionized water in the ratio (10:1) means:
25.0 g of copper sulphate pentahydrate: 2.5g of ferrous heptahydrate were dissolved in 100mL deionized water in ratio (10:1, 25:2.5).
Line 96: 1:1
The mixture solution of copper and ferrous precursors was added slowly to the Lactulose solution in a ratio (1:1) means:
After preparation the mixture of the copper sulphate pentahydrate and ferrous heptahydrate, the Lactulose solution “20 g of lactulose and 50 g of polyethylene glycol 6000 were dissolved in 100 mL deionized water and the pH of the solution was adjusted to alkaline medium to 12.0 and heated to 70° C” was added to this mixture in ratio (1:1, 100 mL of metal salt mixture : 100 mL of lactulose solution ). New section marked in red have been added, lines 91-100 as follows:
“Twenty grams of lactulose and fifty grams of polyethylene glycol 6000 were dissolved in 100 mL deionized water and the pH of the solution was adjusted to alkaline medium to 12.0 and heated to 70° C. Mixture of copper sulfate pentahydrate and ferrous heptahydrate in concentration 27.5 % wt/v dissolved in deionized water in the ratio (10:1) respectively was prepared and heated to 70° C. The mixture solution of copper and ferrous precursors was added slowly to the Lactulose solution in a ratio (1:1) and heated to 70° C for 60 minutes with constant stirring at 1000 rpm. Then the suspension was kept for one day (24 hours) to settle. Subsequently, the centrifugation was implemented for the suspension through 8000 rpm for 5 minutes three times. After that, the precipitate was washed using the deionized water three times also at 8000 rpm via centrifuge. Finally, the separated precipitate was calcinated at 200° C in the oven for two hours.”
14 - I am not sure why the authors use Kelvin (e.g., line 103) in some places in the article. They should be Celsius.
Indeed, Kelvin is the formal usage of the temperature values in the adsorption approaches as it is customary to use temperature units in Kelvin.
15- I cannot follow the methods and materials sections. The authors should write these sections in detail so that others can repeat the experiments.
Many thanks for this comment, that the preparation method was rearranged as the following to be more concise and informative.
“Twenty grams of lactulose and fifty grams of polyethylene glycol 6000 were dissolved in 100 mL deionized water and the pH of the solution was adjusted to alkaline medium to 12.0 and heated to 70° C. Mixture of copper sulfate pentahydrate and ferrous heptahydrate in concentration 27.5 % wt/v dissolved in deionized water in the ratio (10:1) respectively was prepared and heated to 70° C. The mixture solution of copper and ferrous precursors was added slowly to the Lactulose solution in a ratio (1:1) and heated to 70° C for 60 minutes with constant stirring at 1000 rpm. Then the suspension was kept for one day (24 hours) to settle. Subsequently, centrifugation was implemented for the suspension through 8000 rpm for 5 minutes three times. After that, the precipitate was washed using the deionized water three times also at 8000 rpm via centrifuge. Finally, the separated precipitate was calcinated at 200° C in the oven for two hours.”
14 - The authors claim that the red circles in Fig. 3 are cubic particles. As a reader, I cannot see any cubic particles specifically in panels A and D.
We are sorry for the unpleasant reviewing process! We initially uploaded all the figures with a resolution higher than 300 dpi, but they were all compressed in the reviewing manuscript. Now, we again updated all the figures with a higher resolution.
15 - The conclusion should be expanded.
- Done

Reviewer 3 Report
This paper is dedicated to a new nanocomposite based on C-Fe-Cu nanoparticles loaded with known anticancer drug doxorubicin for hepatocellular carcinoma treatment. With undeniable importance of finding new ways in the treatment of cancer, few questions appear concerning methodology and scientific approach presented in the paper.
1. What makes the authors assured that these DOX-loaded C/Fe/Cu nanoparticles target the liver and hepatocellular carcinoma cells? Is there any target drug delivery approach preventing indiscriminate drug release?
2. What precise composition (ratio) of the elements (C, Fe, Cu) comprising these nanoparticles? Is the oxygen also present in the composite (according to FTIR data)?
3. Rows 102 and 104: how can the concentration be 100g/ml and 20 to 100g/ml, while DOX solubility in water is about 10mg/ml?
4. Multiple models to estimate DOX absorption were mentioned in the paper. How they correlate to each other and support spectrophotometric DOX loading determination?
5. Do authors consider “stirring speed” (in rpm) in Design of the experiments and further for loading of the composite with DOX or releasing of DOX? If for DOX release, it is not clear how it may relate to in vitro testing.
Additionally to the questions, the manuscript is written with poor English and organized rather haphazardly. It needs a revision for clear and well-reasoned representation of the results. Scale bars should be clearly marked with values in Fig 3. Reference to “Figure a” should be substituted to corrected Figure number.
Author Response
Dear Editor and Reviewers:
Thank you for your helpful comments concerning our manuscript entitled “A novel C@Fe@Cu Nanocomposite Loaded with Doxorubicin Tailored for the Treatment of Hepatocellular Carcinoma” (pharmaceutics-1850352). These opinions are all valuable and helpful for further revising and improving the quality of the manuscript. We have studied the suggestions carefully and made the corrections accordingly.
The detailed responses to the reviewer’s comments are shown as follows
Reviewer #3: Thank you very much for your suggestions and we really appreciate your efforts
in reviewing our manuscript. We have revised the manuscript accordingly. Our point-by-point
responses are detailed as below.
1- What make the authors assured that these DOX-loaded C/Fe/Cu nanoparticles target the liver and hepatocellular carcinoma cells ? is there any target drug delivery approach preventing indiscriminate drug release?
Cancer cells differ from normal cells in that they divide faster and this requires more nutrients such as glucose and amino acids also,iron and copper are very essential for cancer cells proliferation and metabolism (Jung et al., 2019) (Hsu et al., 2020). combining iron and copper with doxorubicin in one entity Make it coveted for cancer cells and so, selectively delivers high amount of drug to tumour-affected tissue (targeting)which leads to reducing the unwanted effects of the drug in the non-affected ones.
The formulation will be deposited not only in hepatic cancer cells but also any other cancer cells but we tested it on HCC because doxorubicin is very effective against this type of cancer but with a lot of serious side effects such as cardiotoxicity,it is used only by Transarterial chemoembolization or TACE but our formula will make it is possible to be used systematic with less adverse effects.
- What precise composition (ratio) of the elements (C, Fe, Cu) comprising these nanoparticles? Is the oxygen also present in the composite (according to FTIR data)?
Indeed, we could not to make the elemental analysis using ICP-MS. We apologize for not carrying out this experiment. This analysis lab is partially shut down due to the maintenance of the instrument and shortage of the spare parts. We hope that the reviewer understands our challenging situation. But the XRD and SEM mapping analyses revealed and assured the presence of these components. As oxygen, yes the oxygen is present in the composite, where this finding assured the adsorption of the Dox molecules from the function groups through the biosynthesis process.
- Rows 102 and 104: how can the concentration be 100g/ml and 20 to 100g/ml, while DOX solubility in water is about 10mg/ml?
We apologize for this unintended typing error where the right concentration is µg/mL not g/mL “a stock aqueous solution of DOX was prepared at a concentration of 10 mg/mL.” and the sentence corrected as follows:
“A stock aqueous solution of DOX at a concentration of 10 mg/mL was produced at room temperature 293 K followed by three new sets of fresh serial dilution concentrations ranging from 20 to 100 µg/mL (F1).”
Scale bars should be clearly marked with values in Fig 3. Reference to “Figure a” should be substituted to corrected Figure number.
The original figures were included. Unfortunately, the resolution could not be modified. Please, you can use the magnification to more details in the figures.
- Multiple models to estimate DOX absorption were mentioned in the paper. How they correlate to each other and support spectrophotometric DOX loading determination?
- These models were correlated to adjust the optimized formulation parameters that improve LE% and enhance DOX bioavailability.
- Do authors consider “stirring speed” (in rpm) in Design of the experiments and further for loading of the composite with DOX or releasing of DOX? If for DOX release, it is not clear how it may relate to in vitro testing.
- Many thanks for this valuable comment. Indeed, stirring speed is an important factor in adsorption process. We aimed to investigate its effect on the amount of DOX adsorbed at NC surface. This will affect LE%, so we studied it. Also, Stirring speed may affect the adsorption forces between DOX and NC surface, this is because we studied DOX release and investigate stirring speed effect on it.
6- Additionally to the questions, the manuscript is written with poor English and organized rather haphazardly. It needs a revision for clear and well-reasoned representation of the results. Scale bars should be clearly marked with values in Fig 3. Reference to “Figure a” should be substituted to corrected Figure number.
- Thank you for your comments on improving this manuscript. We have thoroughly revised manuscript and checked it by the Grammarly® premium version and cross checked by native English speaker.
We are sorry for the unpleasant reviewing process! We initially uploaded all the figures with aresolution higher than 600 dpi, but they were all compressed in the reviewing manuscript. Now, we again updated all the figures with a higher resolution.
Arora, H. C., Jensen, M. P., Yuan, Y., Wu, A., Vogt, S., Paunesku, T., & Woloschak, G. E. (2012). Nanocarriers Enhance Doxorubicin Uptake in Drug-Resistant Ovarian Cancer CellsNanocarriers Enhance Clathrin-Mediated Endocytosis. Cancer research, 72(3), 769-778.
Cai, W., Guo, M., Weng, X., Zhang, W., & Chen, Z. (2019). Adsorption of doxorubicin hydrochloride on glutaric anhydride functionalized Fe3O4@SiO2 magnetic nanoparticles. Materials Science and Engineering: C, 98, 65-73. https://doi.org/https://doi.org/10.1016/j.msec.2018.12.145
Franco, Y. L., Vaidya, T. R., & Ait-Oudhia, S. (2018). Anticancer and cardio-protective effects of liposomal doxorubicin in the treatment of breast cancer. Breast Cancer: Targets and Therapy, 10, 131.
Hsu, M. Y., Mina, E., Roetto, A., & Porporato, P. E. (2020). Iron: an essential element of cancer metabolism. Cells, 9(12), 2591.
Jung, M., Mertens, C., Tomat, E., & Brüne, B. (2019). Iron as a central player and promising target in cancer progression. International journal of molecular sciences, 20(2), 273.
Karimzadeh, S., Safaei, B., & Jen, T.-C. (2021). Theorical investigation of adsorption mechanism of doxorubicin anticancer drug on the pristine and functionalized single-walled carbon nanotube surface as a drug delivery vehicle: A DFT study. Journal of Molecular Liquids, 322, 114890. https://doi.org/https://doi.org/10.1016/j.molliq.2020.114890
Khan, M. I., Nadeem, I., Majid, A., & Shakil, M. (2021). Adsorption mechanism of Palbociclib anticancer drug on two different functionalized nanotubes as a drug delivery vehicle: A first principle’s study. Applied Surface Science, 546, 149129.
Khasraw, M., Bell, R., & Dang, C. (2012). Epirubicin: Is it like doxorubicin in breast cancer? A clinical review. The Breast, 21(2), 142-149. https://doi.org/https://doi.org/10.1016/j.breast.2011.12.012
Pilco-Ferreto, N., & Calaf, G. M. (2016). Influence of doxorubicin on apoptosis and oxidative stress in breast cancer cell lines. International journal of oncology, 49(2), 753-762.

Round 2
Reviewer 1 Report
The manuscript can be accepted in the revised form.
Reviewer 3 Report
Authors addressed my comments.